# Carbon Dioxide Reduction with Hydrogen on Fe, Co Supported Alumina and Carbon Catalysts under Supercritical Conditions

**DOI:** 10.3390/molecules26102883

**Published:** 2021-05-13

**Authors:** Viktor I. Bogdan, Aleksey E. Koklin, Alexander L. Kustov, Yana A. Pokusaeva, Tatiana V. Bogdan, Leonid M. Kustov

**Affiliations:** 1N.D. Zelinsky Institute of Organic Chemistry, Leninsky Prospect, 47, 119991 Moscow, Russia; akoklin@gmail.com (A.E.K.); kyst@list.ru (A.L.K.); yana_pokusaeva@inbox.ru (Y.A.P.); chemist2014@yandex.ru (T.V.B.); 2Chemistry Department, Lomonosov Moscow State University, Leninskie Gory, 1, Bldg. 3, 119992 Moscow, Russia

**Keywords:** carbon dioxide, carbon monoxide, methane, supercritical CO_2_, iron nanoparticles, cobalt nanoparticles

## Abstract

Reduction of CO_2_ with hydrogen into CO was studied for the first time on alumina-supported Co and Fe catalysts under supercritical conditions with the goal to produce either CO or CH_4_ as the target products. The extremely high selectivity towards methanation close to 100% was found for the Co/Al_2_O_3_ catalyst, whereas the Fe/Al_2_O_3_ system demonstrates a predominance of hydrogenation to CO with noticeable formation of ethane (up to 15%). The space–time yield can be increased by an order of magnitude by using the supercritical conditions as compared to the gas-phase reactions. Differences in the crystallographic phase features of Fe-containing catalysts cause the reverse water gas shift reaction to form carbon monoxide, whereas the reduced iron phases initiate the Fischer–Tropsch reaction to produce a mixture of hydrocarbons. Direct methanation occurs selectively on Co catalysts. No methanol formation was observed on the studied Fe- and Co-containing catalysts.

## 1. Introduction

Carbon dioxide conversion into reduced forms (CO, CH_4_ and CH_3_OH) with the further downstream production of valuable products (aldehydes, acids and liquid fuel) by conventional processes (Fisher–Tropsch synthesis, carbonylation, etc.) is the key step in CO_2_ utilization and sequestration. This approach should be implemented first to capture and utilize the industrial CO_2_ exhausts like metallurgy, heat plant exhausts and municipal wastes incineration. Obviously, the problem of utilization of naturally occurring CO_2_ sources or car exhausts is still very complicated. Among the primary products of CO_2_ reduction, CO, CH_4_ and CH_3_OH deserve attention in views of the further downstream processing [1,2,3,4]. Heterogeneous catalysts are more preferable compared to homogeneous counterparts in terms of stability, separation, handling and reuse, and reactor design, which is transformed into lower costs for large-scale productions [4,5,6].

The methanation reaction is known to proceed quite efficiently on Ni- or Ru-containing catalysts [7]. The catalysts for CO_2_ methanation have been reviewed in [8]. Ru-containing catalysts supported on a ceramic sponge (1 wt % Ru, particle size of 5–20 nm) were developed [9]. 

A. Martin et al. reported on Ni- and Ru-containing catalysts (5 wt %) for CO_2_ hydrogenation at 350–400 °C, 1–20 bar, molar ratio of H_2_/CO_2_ = 4:1, with a gas hourly space velocity 6000 h^−1^ [10]. Ru/ZrO_2_ catalysts demonstrated the highest activity compared to other catalysts studied in this work [11]. The methane yields increased from 70% to 93–96% with increasing the pressure of the reaction mixture from 1 to 20 bar, with methane being the only product. A RuNi/ZrO_2_ catalyst was reported [12] to reach a 100% CO_2_ conversion to methane at 300–400 °C and space velocities up to 36,000 h^−1^. Noteworthy that the metal loading in the most active catalysts was quite significant (3–5 wt %) and the gas mixture was significantly diluted with H_2_ and N_2_ (a 4–7 times excess over CO_2_). The maximum space–time yield of methane formation reached 40–100 g/g(metal) h.

The recent data also demonstrated that bimetallic catalysts (Ru-Ni/CeO_2_-ZrO_2_) exhibit enhanced performance in methane formation [13]. Kwak et al. [14] studied the role of the Ru particle size in the catalytic performance of Ru/Al_2_O_3_.

The catalytic conversion of CO_2_ to CO via the reverse water–gas-shift reaction (RWGSR) has been generally considered as one of the most economically viable processes for CO_2_ conversions [4,15].
CO_2_ + H_2_ = CO + H_2_O, ΔH_298_ = 41.2 kJ mol^−1^

Diverse metals exhibit the catalytic activity in RWGSR, with Cu, Au and Ag deserving most attention [16,17]. The studies reviewed included both homogeneous and heterogeneous catalysts and electrocatalytic or photoelectrocatalytic conversion [18,19,20,21]. 

Though the methanol synthesis from CO_2_ and H_2_ is a separate area and the best studied system is a CuO-ZnO catalyst commonly used for methanol synthesis from CO and H_2_, we can mention here the recent review by I. Ganesh [22] and an interesting work related to In_2_O_3_ [23].

In spite of the wide coverage of the research related to CO_2_ hydrogenation, the use of supercritical CO_2_ (scCO_2_) in the CO_2_ hydrogenation reaction was the focus of only a few recent publications [24,25,26,27]. However, the target product was formic acid or formamide, which determined the choice of the catalysts (homogeneous Ru complexes) and the mode of operation. There is a number of our works devoted to the hydrogenation of CO_2_ under supercritical conditions [28,29,30,31,32,33,34]. A supercritical media may enhance the catalyst activity and prolong its lifetime [34].

Obviously, the choice of non-noble catalysts to replace Ru, Au and other expensive components in the catalysts of CO_2_ hydrogenation would be a step forward in the development of robust CO_2_ utilization catalytic systems. On the other hand, the use of supercritical conditions for this particular reaction has not been studied in sufficient detail. The goal of this work was to fill this gap by exploring rather simple and non-expensive catalysts containing iron and cobalt on an accessible commercial carrier (alumina), i.e., Fe/Al_2_O_3_ and Co/Al_2_O_3_ heterogeneous catalysts, with CO_2_ being both the reagent and the supercritical medium in a flow reactor.

## 2. Results and Discussion

In a general case, the interaction between CO_2_ and H_2_ occurs with the formation of several products, with CO (1) and further formed hydrocarbons (2), methane (3) and methanol (4) being the main products:CO_2_ + H_2_ → CO + H_2_O(1)
*x*CO + (*x* + *y*/2)H_2_ → C_x_H_y_ + *x*H_2_O(2)
CO_2_ + 4H_2_ → CH_4_ + 2H_2_O(3)
CO_2_ + 3H_2_ → CH_3_OH + H_2_O(4)

The catalytic data on the conversion and selectivities to the main products on the Fe- and Co-supported alumina and carbon catalysts are presented in Table 1.

Comparison of the performance of the Fe and Co nanoparticles in the CO_2_ hydrogenation under supercritical conditions demonstrates a significant difference in the nature of products formed. Whereas the Fe nanoparticles are not quite selective to any specific product and CO, CH_4_ and C_x_H_y_ are formed, the Co nanoparticles produce methane with a selectivity of 96–98% with the rest being ethane, but not CO. Noteworthy that the Co catalyst is more active than the Fe catalyst. Its performance in the low-temperature region (200–350 °C) is comparable to that of the Fe-based catalysts at 400 °C (Figure 1). An interesting result is the formation of C_1_–C_10_ hydrocarbons with a selectivity up to 60%, though at low CO_2_ conversions. It was also of interest to compare the performances of Fe-containing catalysts on different carriers, with alumina and carbon having been chosen for the comparative purposes.

The concentration of hydrogen at the reactor outlet is about 1.5–2% for the Co-catalyst, while it remains at the level of 33–35% for the Fe catalyst because of the different stoichiometry of the reactions occurring on these two catalysts (methane formation on the Co/Al_2_O_3_ catalyst and predominant CO formation on the Fe/Al_2_O_3_ catalyst). The stability of the operation was studied for both catalysts at 400 °C. The activity of both catalysts was kept quite constant within 7 h.

The advantage of the catalytic CO_2_ hydrogenation performed under supercritical conditions is the high throughput, i.e., the productivity of the catalyst expressed in terms of grams of CO_2_ passed or converted per gram of the catalyst per hour. The gas-phase tests described in the literature were performed in diluted gas mixtures, with H_2_ or N_2_ serving as diluents, including one of the best results reported by A. Martin et al. who tested the Ni- and Ru-containing catalysts (5 wt %) in CO_2_ reduction with H_2_ at 350–400 °C, 1–20 bar, molar ratio of H_2_/CO_2_ = 4:1, with a gas hourly space velocity 6000 h^−1^ [10,11,12]. Additionally, the productivity was quite limited (about 0.2 g/g·h). In our tests, the productivity reached 1.54 g/g h or 0.4 mol CO_2_ per 1 g of metal per hour, which is about 7–8 times higher compared to the literature data.

The following phase transformations can occur on the surface of Fe-containing catalysts in a reducing medium: Fe_2_O_3_ → Fe_3_O_4_ → FeO → Fe. The TEM method indicates the formation of mixed iron oxide Fe_3_O_4_ with the structure of reverse spinel in the initial samples on Sibunit and aluminum oxide (Figure 2). The formation of phases of the structural type of spinel contributes to the selective formation of CO in the water gas shift reaction at 350 °C. For the Al_2_O_3_ carrier, the γ-alumina with the spinel structure is also known, so this carrier stabilizes the spinel phase of the Fe_3_O_4_ type, which prevents further reduction of iron. The use of a carbon carrier promotes the reduction of iron, which leads to the deeper reduction of the Fe_3_O_4_ phase and the formation of carbide phases. It should be noted that the mixed cobalt oxide Co_3_O_4_ is characterized by the phase of normal spinel. Perhaps this somehow affects the catalyst performance in the reactions. In normal spinel, Co^+3^ ions are located in an octahedral environment of oxygen atoms, and Co^+2^ ions are located in a tetrahedral environment. In the reverse spinel, Fe^+2^ ions are in the octahedral environment, and Fe^+3^ ions are localized in both the octahedral and tetrahedral environments. The latter determines the ferromagnetic properties of magnetite.

Differences in the crystallographic phase features affected the catalytic results (Table 1). The presence of reverse spinel oxide phases in Fe-containing catalysts caused the water gas shift reaction to form carbon monoxide, while the reduced iron phases initiated the Fischer–Tropsch reaction to produce a mixture of hydrocarbons. Direct methanation occurs selectively on Co deposited catalysts. Noteworthy that the variation of the carrier (alumina or carbon) in the case of Fe-containing catalysts did not show any significant difference in the performance of the catalysts in terms of either CO_2_ conversion or selectivity to main products (CO, CH_4_ or C2+ hydrocarbons). Since Fe/Al_2_O_3_ catalyst demonstrated the best performance in CO formation compared to Co/Al_2_O_3_ and Fe/C catalysts, this catalyst was studied in more detail by using diverse physicochemical methods of characterization. The better performance of the Fe/Al_2_O_3_ catalyst compared to the Fe/C catalyst is illustrated by Figure 1 presenting the productivity in terms of the number of moles of CO_2_ converted per 1 g of metal per hour. The Fe/Al_2_O_3_ catalyst outperformed the Fe/C catalyst in the temperature range higher than 350 °C, if one considers CO as a desirable product.

Temperature-programmed reduction of the Fe/Al_2_O_3_ catalyst in comparison with the reduction of the pure Fe_2_O_3_ phase (Figure 3) revealed the first reduction peak at 410 °C due to the conversion of Fe_2_O_3_ into Fe_3_O_4_. This peak was shifted by about 50 °C to higher temperatures compared to the bulk Fe_2_O_3_ phase, as a result of Fe_2_O_3_ interaction with Al_2_O_3_ and stabilization. The second peak at about 620–630 °C can be attributed to the further reduction of Fe_3_O_4_ to FeO. This peak was also shifted toward higher temperatures in comparison with the unsupported bulk sample as a result of stabilization by the alumina carrier. The complete reduction of iron in the supported catalyst to Fe^0^ occurred at about 850 °C. The H_2_/Fe ratio for this catalyst was 0.83 mol/mol, which roughly coincided with the value determined for the bulk oxide (0.86 mol/mol).

The photo of the thin cut of the catalyst grain (Figure 4) shows a very uniform distribution of iron along the grain.

Further information about the state of iron in the Fe/Al_2_O_3_ catalyst was derived by diffuse-reflectance FTIR spectroscopy using CO as a probe molecule. The spectrum of CO adsorbed on this catalyst is presented in Figure 5. The spectrum measured before catalysis demonstrates the band at 2191 cm^−1^ that can be assigned to complexes with Fe^3+^ ions or, possibly to CO adsorbed on the low-coordinated ions of the carrier. The main band at 2111 cm^−1^ with two small shoulders at about 2145 cm^−1^ and 2068 cm^−1^ can be ascribed to different reduced forms of iron, including Fe^2+^ and charged Fe^0^ species. Comparison of the spectra measured before and after catalysis shows that the sample after catalysis was characterized by the presence of more reduced forms of iron (the band at 2109 cm^−1^), although the integral intensity of this band (2109–2111 cm^−1^) decreased insignificantly for the catalyst after the reaction compared to the fresh catalyst.

The TEM photo of the catalyst presented in Figure 6 demonstrates that the size of iron oxide particles was about 5 nm. EDX analysis confirmed the uniform distribution of iron over the surface of the catalyst.

The XPS study of the supported iron on alumina (Figure 7) showed that iron existed in a mixture of Fe^3+^ and Fe^2+^ with roughly equal contributions.

Thus, different physicochemical methods provide the information about the presence of several forms of iron on the surface of the catalyst. With account of the H_2_-TPR and XPS data, one can conclude that a mixture of Fe^3+^ and Fe^2+^ oxide species is found in the Fe-catalysts in the temperature range under study (200–500 °C), thus providing a possibility of occurrence of Fischer–Tropsch synthesis reaction yielding C2+ hydrocarbons. Under the reaction conditions, most probably, most of these forms are converted into an iron carbide, which is considered in the literature to be the main state of supported iron in the course of the reverse water–gas shift reaction. This explains a significant (up to 16–27%) contribution of Fisher–Tropsch synthesis process in the overall pattern of reactions occurring during CO_2_ hydrogenation on iron-containing catalysts. On the contrary, the Co-containing catalyst demonstrates a very high (about 98%) selectivity toward methane, which is a less interesting option of the CO_2_ hydrogenation.

## 3. Materials and Methods

The Co/Al_2_O_3_, Fe/Al_2_O_3_ and Fe/C catalysts were prepared by incipient wetness impregnation of γ-Al_2_O_3_ (surface area, 270 m^2^/g) or synthetic carbon material Sibunit (surface area, 340 m^2^/g) using aqueous solutions of the corresponding nitrate salts (Acros Organics, Fair Lawn, NJ, USA, 99+%). The metal loading was 5 wt %. The alumina-based catalysts were calcined in a flow of air at 500 °C for 5 h, then in a flow of hydrogen at 500 °C for 5 h. The Fe/C catalyst were calcined in a flow of argon at 450 °C for 4 h, then in a flow of hydrogen at 400 °C for 2 h. The metal dispersion determined by oxygen titration was 42% for Fe/Al_2_O_3_ and 47% for Co/Al_2_O_3_. The average metal particle size determined from TEM measurements was 12.3 nm for Fe/Al_2_O_3_ and 11.0 nm for Co/Al_2_O_3_.

The reaction of CO_2_ with H_2_ was studied under supercritical conditions in a fixed-bed stainless steel reactor. The catalyst loading in the reactor was 1 g. The reaction temperature was ranged from 200 to 500 °C. Carbon dioxide was supplied with a syringe pump under the pressure of 80 atm, hydrogen was fed via a mass flow controller. The H_2_:CO_2_ ratio was equal to 1:1, gas hourly space velocity (GHSV) was 4800 NmLg_cat_^−1^h^−1^. The pressure in the reactor was maintained at 80 atm using a back pressure valve. Analysis of products was performed with a Crystal 5000.2 gas chromatograph (Chromatek, Yoshkar-Ola, Russia) with a thermal conductivity detector and Porapak Q and zeolite CaA packed columns. The carbon balance was closed at 99–100%.

The composition and surface structure of the studied catalysts were determined by transmission electron microscopy (TEM) using a JEOL-2100F electron microscope (JEOL, Tokyo, Japan) in the light and dark field modes at an accelerating voltage of 200 kV. Elemental analysis of the surface with deposited metal particles was carried out using the energy-dispersive X-ray spectroscopy (EDX) method using the TEM attachment.

## 4. Conclusions

Thus, Fe/alumina, Co/alumina and Co/C catalysts were studied in reverse water gas shift reaction in supercritical CO_2_ for the first time. The application of supercritical CO_2_ in the CO_2_ hydrogenation reaction turns out to be an efficient approach to enhance the space–time yield of the products (CO for supported Fe/alumina and Fe/C catalysts or CH_4_ for the Co-based catalyst). The Co/alumina and Fe/alumina catalysts revealed a rather high activity in CO_2_ hydrogenation, with the highest selectivity toward CH_4_ (98%) revealed by the Co catalyst. The productivity reached 1.54 g/g h, which is about 7–8 times higher compared to the literature data. A mixture of Fe^3+^ and Fe^2+^ oxide species was found in the Fe-catalysts, which may be further transformed partially into carbide moieties under the reaction conditions thus providing a possibility of occurrence of Fischer–Tropsch synthesis reaction yielding C2+ hydrocarbons. Thus, the use of non-noble (Co and Fe) and cheap metals (Fe) allows one to solve the problem of utilization of critical metals.

## Figures and Tables

**Figure 1 molecules-26-02883-f001:**
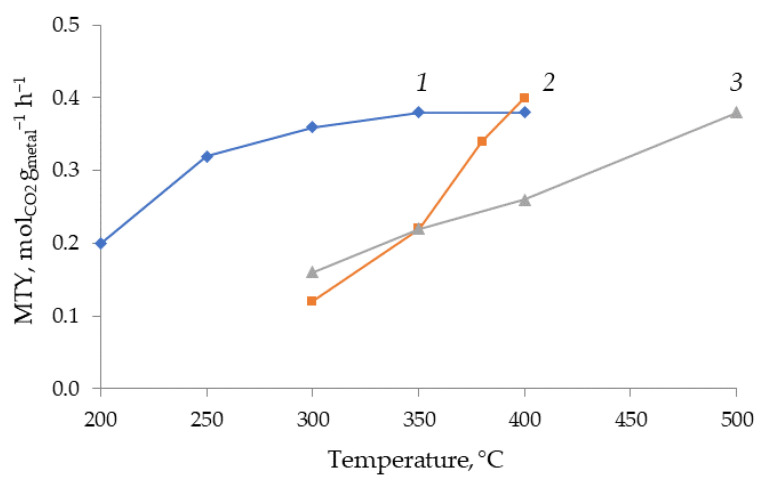
Performance of Co/Al_2_O_3_ (1), Fe/Al_2_O_3_ (2) and Fe/C (3) catalysts in CO_2_ reduction with H_2_ (H_2_:CO_2_ = 1:1, 80 atm, GHSV = 4800 NmLg_cat_^−1^h^−1^). MTY (metal-time yield) was calculated based on CO_2_ conversion, gas flow, and cobalt or iron content.

**Figure 2 molecules-26-02883-f002:**
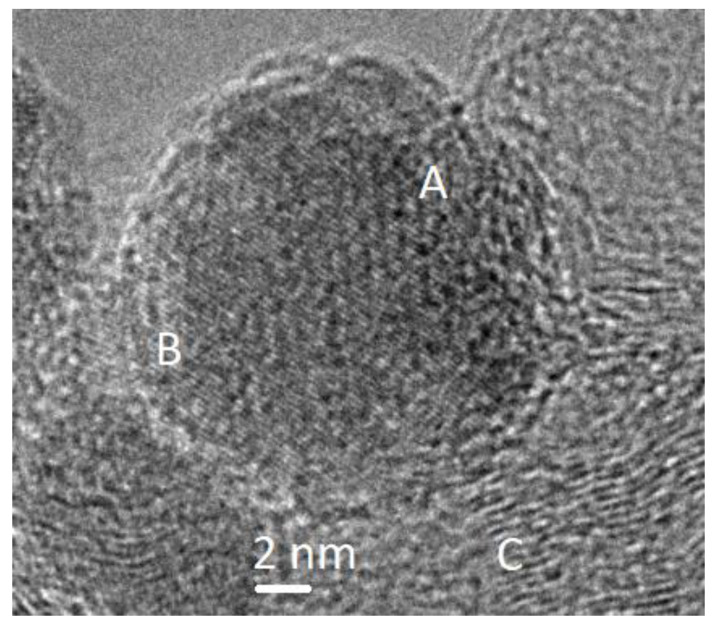
A typical view of the Fe/C catalyst surface: (**A**)—magnetite Fe_3_O_4_, (**B**)—FeO and (**C**)—partially graphitized Sibunit carrier.

**Figure 3 molecules-26-02883-f003:**
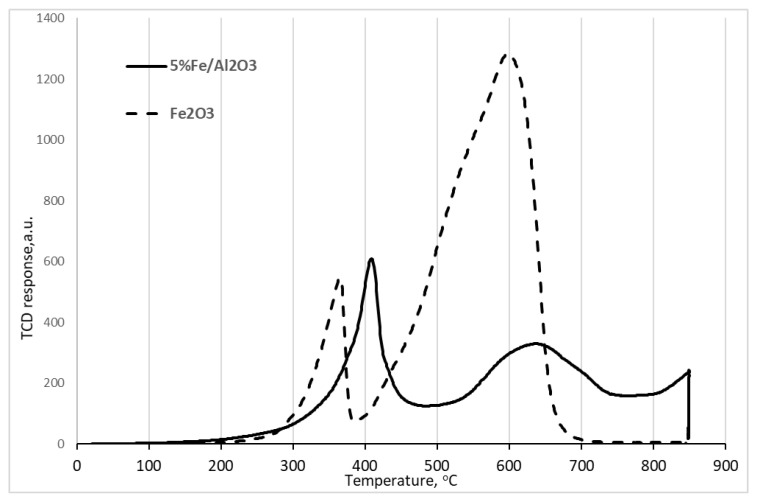
TPR curves for the reduction of bulk Fe_2_O_3_ and supported Fe/Al_2_O_3_.

**Figure 4 molecules-26-02883-f004:**
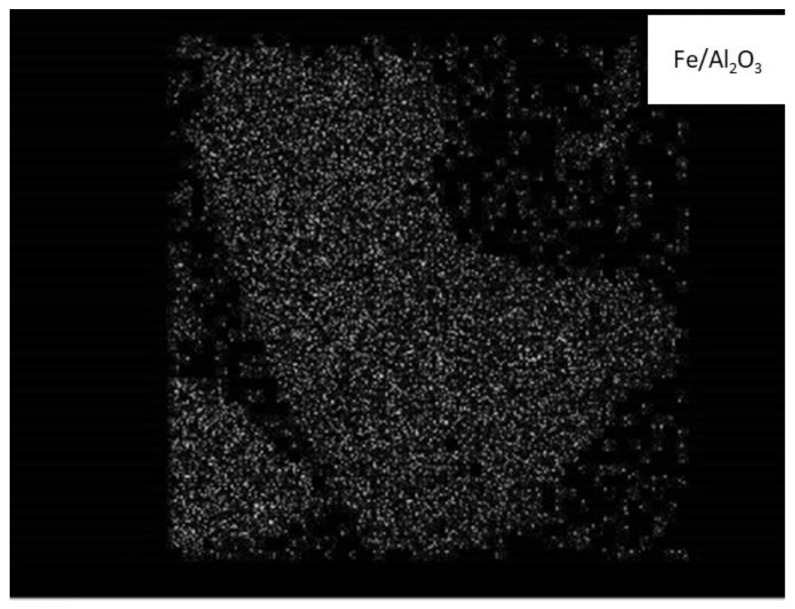
Iron distribution on the surface of the polished cut of the grain of the 5%Fe/Al_2_O_3_ catalyst.

**Figure 5 molecules-26-02883-f005:**
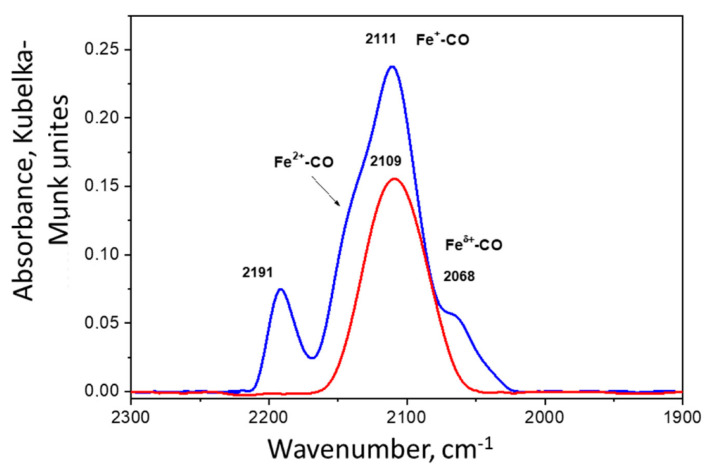
Diffuse-reflectance FTIR spectra of CO adsorbed on the Fe/Al_2_O_3_ catalyst before catalysis (blue) and after catalysis (red).

**Figure 6 molecules-26-02883-f006:**
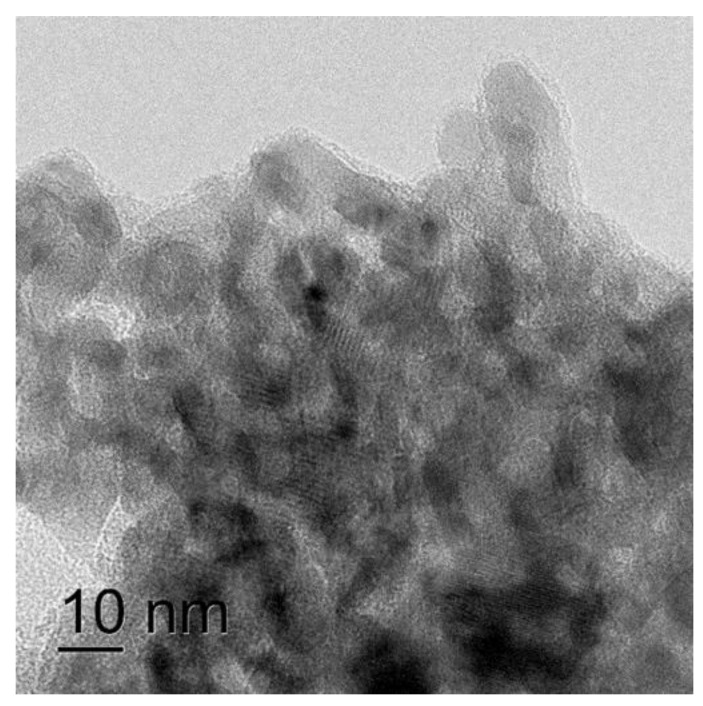
TEM photo of the Fe/Al_2_O_3_ catalyst.

**Figure 7 molecules-26-02883-f007:**
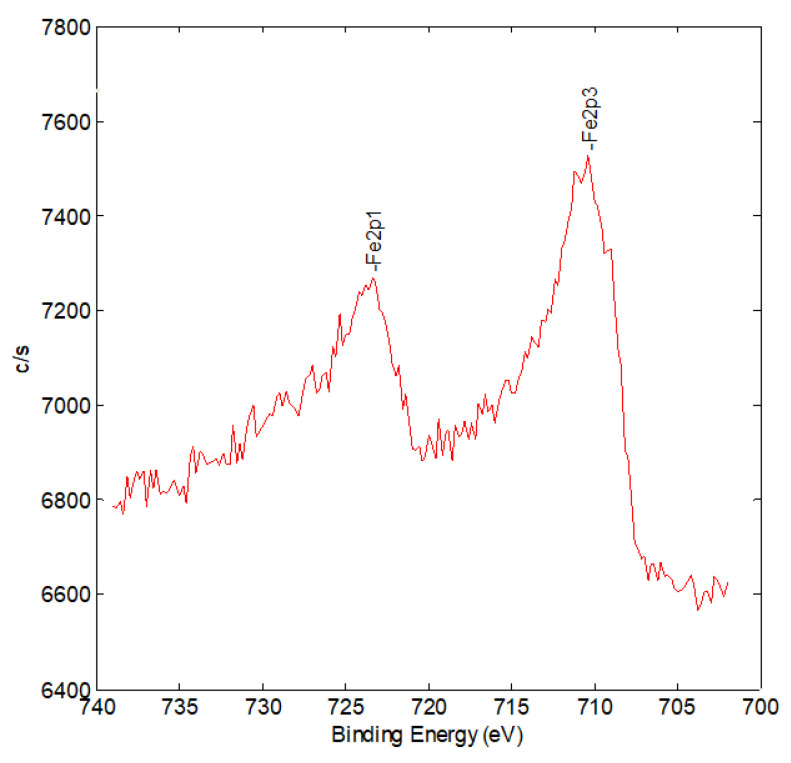
XPS spectrum of the Fe line for the 5%Fe/Al_2_O_3_ catalyst after catalysis.

**Table 1 molecules-26-02883-t001:** Performance of the Fe- and Co- supported alumina and carbon catalysts in CO_2_ reduction with H_2_ (H_2_:CO_2_ = 1:1, 80 atm, GHSV = 4800 NmLg_cat_^−1^h^−1^).

Catalyst	*T*, °C	CO_2_ Conversion, %	Selectivity, vol. %		
			CO	CH_4_	C_x_H_y_ *
Co/Al_2_O_3_	200	10	0	96	4
	250	16	0	96	4
	300	18	0	98	2
	350	19	0	98	2
	400	19	0	98	2
Fe/Al_2_O_3_	250	0	-	-	-
	300	6	45	40	15
	320	7	45	43	11
	350	11	40	44	16
	380	17	60	29	11
	400	20	66	25	9
Fe/C	300	8	62	21	17
	350	11	58	15	27
	400	13	73	10	17
	500	19	90	6	4

* Ethane on Co/Al_2_O_3_ and C_2_–C_12_ hydrocarbons on Fe-based catalysts.

## Data Availability

The data presented in this study are available on request from the corresponding author.

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
