# Peer review of "Carbon Dioxide Reduction with Hydrogen on Fe, Co Supported Alumina and Carbon Catalysts under Supercritical Conditions"

_molecules, 2021, doi:10.3390/molecules26102883_

Round 1
Reviewer 1 Report
Review of "Carbon Dioxide Reduction with Hydrogen on Fe, Co Supported Alumina and Carbon Catalysts under Supercritical Conditions" by Viktor I. Bogdan et al.
The manuscript by Viktor I. Bogdan et al. reports on the synthesis and testing of Fe- and Co-based catalysts for the catalytic hydrogenation of supercritical CO2 (scCO2). While the authors claim that using scCO2 for the CO2 hydrogenation reaction is a “new approach”, this has already been extensively reported (e.g. refs 27-38), even by the same authors (e.g. ref 28-34, 38 – inappropriate redundant self-citations may be also noticed). The use of Fe and Co catalysts under scCO2 conditions does not consist of enough novelty for publication, especially considered the almost absent characterization results reported.
I therefore think the manuscript is not suitable for scientific publication. My comments follow.
1. Reaction 1 is not properly balanced. It should be just one (not two) CO molecule.
2. The scheme in section 2 should be (if at all) reported in the introduction, and should be either modified, or reported as reproduced, as I noticed it was already included in ref. 28 by the same authors (self-plagiarism).
3. A CO2:H2 ratio of 1:4 should also be explored for Co, where methanation was the main observed reaction (the reaction requires 4 hydrogens per CO2). Moreover, since the moles in the methanation reaction go from 5 to 3 at complete conversion, the effect of conversion on the reactor pressure and supercritical conditions should be discussed.
4. Table 1 is not per se needed, since the results are reported in Tables 2 and 3.
5. Table 2 and 3 should be merged.
6. Figure 1 should be adapted to compare the three catalysts studied.
7. The actual active phase for Fe and Co catalysts under CO2 hydrogenation conditions is highly debated, and the role of the different phases is uncertain. Operando spectroscopy techniques are necessary to elucidate relations between catalytic performance and chemical composition/ structure. The authors fail to discuss this very important issue, and completely overlook the possible role of Co and Fe carbide species (e.g. A. V Puga, On the nature of active phases and sites in CO and CO2 hydrogenation catalysts, Catal. Sci. Technol. 8 (2018) 5681–5707. doi:10.1039/C8CY01216D). Overall, a more critical and detailed discussion should be included when discussing Fe (and Co) phases.
8. Magnetometry is mentioned on page 5, but results are not reported.
9. The (fresh and used) catalysts should be characterized much more in depth: for example, by running routine characterization by N2 physisorption, XRD (for crystal phases), ICP (metal content), DRIFTS or Raman (surface species, oxide phases, etc.). TEM analysis results should also be reported, including Fe and Co/Al2O3, and proper nanoparticles size analysis, especially since the CO2 hydrogenation reaction is known to be structure sensitive (i.e. activity depends on NPs size).
10. The authors should report the purity of all reagents used in section 3 for the sake of reproducibility.
Author Response
Manuscript ID: molecules-1150813
Title: Carbon Dioxide Reduction with Hydrogen on Fe, Co Supported Alumina and Carbon Catalysts under Supercritical Conditions
Dear Editor and Reviewers,
We would like to thank you for the careful reading of our manuscript and your questions, remarks, and recommendations. We have revised and expanded the manuscript and we kindly ask you to review the revised version. Please find below our responses to the reviewers comments. All significant changes in the manuscript are highlighted in yellow. Also, we checked the use of English and made corrections in the text (not shown in the revised manuscript).
Responses to Reviewer #1 Comments
Reviewer #1: The manuscript by Viktor I. Bogdan et al. reports on the synthesis and testing of Fe- and Co-based catalysts for the catalytic hydrogenation of supercritical CO2 (scCO2). While the authors claim that using scCO2 for the CO2 hydrogenation reaction is a “new approach”, this has already been extensively reported (e.g. refs 27-38), even by the same authors (e.g. ref 28-34, 38 – inappropriate redundant self-citations may be also noticed). The use of Fe and Co catalysts under scCO2 conditions does not consist of enough novelty for publication, especially considered the almost absent characterization results reported.
Response: The first publications on CO2 hydrogenation under supercritical conditions appeared in 2016 and these are mostly the publications of the authors, therefore, it is still a new approach (5 years is a short period). Also, we removed 3 publications of the authors from the list to reduce self-citation, although, these citations are very much relevant to the subject of the paper and cannot be qualified as redundant. Fe and Co nanoparticles on alumina were studied indeed for the first time under supercritical conditions of CO2 hydrogenation and these results do constitute a novelty of the publication. More characterization data have been added in the manuscript (highlighted text in pages 4-7), including TEM, SEM, EDX, TPR, XPS, and DRIFTS with CO used as a probe molecule.
Reviewer #1: 1. Reaction 1 is not properly balanced. It should be just one (not two) CO molecule.
Response: Sorry about this misprint. It is corrected.
Reviewer #1: 2. The scheme in section 2 should be (if at all) reported in the introduction, and should be either modified, or reported as reproduced, as I noticed it was already included in ref. 28 by the same authors (self-plagiarism).
Response: The scheme was removed. Instead, we used equations of chemical reactions to show the main processes occurring in the course of CO2 hydrogenation, including Fischer-Tropsch synthesis and methanation, as well as methanol formation.
Reviewer #1: 3. A CO2:H2 ratio of 1:4 should also be explored for Co, where methanation was the main observed reaction (the reaction requires 4 hydrogens per CO2). Moreover, since the moles in the methanation reaction go from 5 to 3 at complete conversion, the effect of conversion on the reactor pressure and supercritical conditions should be discussed.
Response: Hydrogenation of carbon dioxide on Co and Fe-containing catalysts was studied under identical conditions with CO being the primary target product, but not CH4. Therefore, the choice was made in favor of Fe-containing catalysts. Despite the development of methods for producing hydrogen and hydrogen energy, hydrogen is still an expensive compound. In this regard, it is of interest to conduct the process with a limited hydrogen content in the mixture.
Reviewer #1: 4. Table 1 is not per se needed, since the results are reported in Tables 2 and 3.
Response: We revised tables significantly. Table 1 was removed, and Tables 2 and 3 were merged into one table.
Reviewer #1: 5. Table 2 and 3 should be merged.
Response: Tables 2 and 3 were merged into one table.
Reviewer #1: 6. Figure 1 should be adapted to compare the three catalysts studied.
Response: We presented the data in a different manner. Now Figure 1 shows the productivities (space-time yields) vs. temperature. The catalysts can be compared now.
Reviewer #1: 7. The actual active phase for Fe and Co catalysts under CO2 hydrogenation conditions is highly debated, and the role of the different phases is uncertain. Operando spectroscopy techniques are necessary to elucidate relations between catalytic performance and chemical composition/ structure. The authors fail to discuss this very important issue, and completely overlook the possible role of Co and Fe carbide species (e.g. A. V Puga, On the nature of active phases and sites in CO and CO2 hydrogenation catalysts, Catal. Sci. Technol. 8 (2018) 5681–5707. doi:10.1039/C8CY01216D). Overall, a more critical and detailed discussion should be included when discussing Fe (and Co) phases.
Response: Unfortunately, we do not have an access to operando spectroscopic method to elucidate the roles of different phases in the catalytic processes occurring in the CO2+H2 system. However, we additionally studied the most efficient catalyst, Fe/Al2O3, by different physicochemical methods. More characterization data have been added in the manuscript (highlighted text in pages 4-7), including TEM, SEM, EDX, TPR, XPS, and DRIFTS with CO used as a probe molecule.
Reviewer #1: 8. Magnetometry is mentioned on page 5, but results are not reported.
Response: Sorry about this misprint. Magnetometry was not used in this study.
Reviewer #1: 9. The (fresh and used) catalysts should be characterized much more in depth: for example, by running routine characterization by N2 physisorption, XRD (for crystal phases), ICP (metal content), DRIFTS or Raman (surface species, oxide phases, etc.). TEM analysis results should also be reported, including Fe and Co/Al2O3, and proper nanoparticles size analysis, especially since the CO2 hydrogenation reaction is known to be structure sensitive (i.e. activity depends on NPs size).
Response: We have focused on Fe/Al2O3 catalyst and studied this catalyst using available physicochemical methods. Characterization data have been added in the manuscript (highlighted text in pages 4-7), including TEM, SEM, EDX, TPR, XPS, and DRIFTS with CO used as a probe molecule. Comparison of the Fe/alumina samples before and after catalysis was made using DRIFTS data with CO as a probe.
Reviewer #1: 10. The authors should report the purity of all reagents used in section 3 for the sake of reproducibility.
Response: Cobalt and iron nitrates used for the preparation of catalysts had the purity of 99+% (Acros Organics) and this information is included in Materials and Methods section.
Reviewer 2 Report
The manuscript deals with the effect of supercritical reaction condition on the CO2 hydrogenation over supported transition metals catalysts. Modifications are required before acceptance, according to the comments below.
- The positive effect of supercritical conditions is questionable and it seems due to the intensification related to the increased pressure rather than to a positive effect on the reaction kinetics. On Fe-based catalysts, CO2 conversions are quite low independently from the support and the reaction conditions. Comparison with literature results is reported but with only few papers. A table with performance of literature and presented results is suggested.
- Page 3, lines 90-92. “Noteworthy that the Co catalyst is more active 90 than the Fe catalyst in the low-temperature region (200-350°C), whereas their activities are 91 comparable at 400°C.” Co-based catalyst appear more active than Fe ones in the whole temperature range; as a matter of fact, CO2 conversion is limited by the almost complete H2 conversion. Due to the formation of CH4, H2 consumption is 4 times higher than that of CO2.
- Why are table 2 and table 3 different? It is unusual to report concentrations for two samples and conversion and selectivity for another one. Please choose one way to report catalytic performance. Moreover, you can report all the data in a single table.
- Equation 1 is wrongly balanced.
Author Response
Manuscript ID: molecules-1150813
Title: Carbon Dioxide Reduction with Hydrogen on Fe, Co Supported Alumina and Carbon Catalysts under Supercritical Conditions
Dear Editor and Reviewers,
We would like to thank you for the careful reading of our manuscript and your questions, remarks, and recommendations. We have revised and expanded the manuscript and we kindly ask you to review the revised version. Please find below our responses to the reviewers comments. All significant changes in the manuscript are highlighted in yellow. Also, we checked the use of English and made corrections in the text (not shown in the revised manuscript).
Responses to Reviewer #2 Comments
Reviewer #2: The positive effect of supercritical conditions is questionable and it seems due to the intensification related to the increased pressure rather than to a positive effect on the reaction kinetics. On Fe-based catalysts, CO2 conversions are quite low independently from the support and the reaction conditions. Comparison with literature results is reported but with only few papers. A table with performance of literature and presented results is suggested.
Response: We included the best performing catalysts in the literature review in the introduction. Now we extended the discussion of the comparable catalysts. However, none of the available (in the literature) Fe-catalysts have been tested in supercritical conditions, therefore, strictly speaking, the comparison is not possible. Thus, we decided that a table proposed by the Reviewer would be misleading.
Reviewer #2: Page 3, lines 90-92. “Noteworthy that the Co catalyst is more active than the Fe catalyst in the low-temperature region (200-350°C), whereas their activities are comparable at 400°C.” Co-based catalyst appear more active than Fe ones in the whole temperature range; as a matter of fact, CO2 conversion is limited by the almost complete H2 conversion. Due to the formation of CH4, H2 consumption is 4 times higher than that of CO2.
Response: We agree with this comment. This sentence was replaced by the following one: “Noteworthy that the Co catalyst is more active than the Fe catalyst: its performance in the low-temperature region (200-350°C) is comparable to that of the Fe-based catalysts at 400°C (Fig. 1).”
Reviewer #2: Why are table 2 and table 3 different? It is unusual to report concentrations for two samples and conversion and selectivity for another one. Please choose one way to report catalytic performance. Moreover, you can report all the data in a single table.
Response: We revised tables significantly. Tables 2 and 3 were merged into one table.
Reviewer #2: Equation 1 is wrongly balanced.
Response: Sorry about this fault, it is corrected.
Reviewer 3 Report
In this paper, the authors report the comparison of Fe and Co supported on Alumina catalysts for CO2 hydrogenation under supercritical conditions. However, they also include an Fe/C catalyst that is not compared in all sections. Although the topic is of high interest, there is a lack of justification and a more detailed discussion would be recommended. Therefore, I would suggest the publication of the manuscript after taking care of the following items:
- The structure of the introduction is not clear. The authors indicate the different processes that take place due to the combination of CO2 and H2, but the reason for the study of Fe and Co catalysts supported on alumina is not clear. In addition, the sentences from 39 to 44 are identical to the introduction in reference number 28. Moreover, there are 11 references of the same authors without including enough information that should be revised.
- The scheme of the main reactions is also similar to that presented in reference 28 and is not part of the results obtained in this work.
- The authors should improve the discussion of the results. There are only three references in the whole section and, besides, it is the same sentence as in the introduction section.
- The authors include three tables, two of them for alumina supported catalysts and one for Fe/C catalyst, but they do not explain the data shown in table 3 for Fe/C material. Can the authors explain the reason to include this data? They could include a section to compare Fe supports.
- Figure 1 and Table 2 contain the same data, which is redundant and should be reorganized.
- In order to explain the species involved in the process, the include only one TEM image of Fe/C that, theoretically, it is not the catalyst of interest in this contribution. Could the authors include the images for alumina supported catalysts?
- Characterization techniques could be explained before providing the data obtained in the experimental section.
Author Response
Manuscript ID: molecules-1150813
Title: Carbon Dioxide Reduction with Hydrogen on Fe, Co Supported Alumina and Carbon Catalysts under Supercritical Conditions
Dear Editor and Reviewers,
We would like to thank you for the careful reading of our manuscript and your questions, remarks, and recommendations. We have revised and expanded the manuscript and we kindly ask you to review the revised version. Please find below our responses to the reviewers comments. All significant changes in the manuscript are highlighted in yellow. Also, we checked the use of English and made corrections in the text (not shown in the revised manuscript).
Responses to Reviewer #3 Comments
Reviewer #3: In this paper, the authors report the comparison of Fe and Co supported on Alumina catalysts for CO2 hydrogenation under supercritical conditions. However, they also include an Fe/C catalyst that is not compared in all sections. Although the topic is of high interest, there is a lack of justification and a more detailed discussion would be recommended. Therefore, I would suggest the publication of the manuscript after taking care of the following items:
The structure of the introduction is not clear. The authors indicate the different processes that take place due to the combination of CO2 and H2, but the reason for the study of Fe and Co catalysts supported on alumina is not clear. In addition, the sentences from 39 to 44 are identical to the introduction in reference number 28. Moreover, there are 11 references of the same authors without including enough information that should be revised.
Response: The first publications on CO2 hydrogenation under supercritical conditions appeared in 2016 and these are mostly the publications of the authors. However, we removed 3 publications of the authors from the list to reduce self-citation, although, these citations are very much relevant to the subject of the paper.
The text (lines 39-44) was substantially revised as follows:
“Ru/ZrO2 catalysts demonstrated the highest activity compared to other catalysts studied in this work [11]. The methane yields increased from ~70% to 93-96% with increasing the pressure of the reaction mixture from 1 to 20 bar, with methane being the only product A RuNi/ZrO2 catalyst was reported [12] to reach a 100% CO2 conversion to methane at 300–400°C and space velocities up to 36 000 h-1. Noteworthy that the metal loading in the most active catalysts was quite significant (3-5 wt. %) and the gas mixture was significantly diluted with H2 and N2 (a 4-7 times excess over CO2).”
Also, the goal of the study was formulated more clearly as follows:
“Obviously, the choice of non-noble catalysts to replace Ru, Au and other expensive components in the catalysts of CO2 hydrogenation would be a step forward in the development of robust CO2 utilization catalytic systems. On the other hand, the use of supercritical conditions for this particular reaction has not been studied in sufficient detail. The goal of this work was to fill this gap by exploring rather simple and non-expensive catalysts containing iron and cobalt on an accessible commercial carrier (alumina), i.e. Fe/Al2O3 and Co/Al2O3 heterogeneous catalysts, with CO2 being both the reagent and the supercritical medium in a flow reactor.”
Reviewer #3: The scheme of the main reactions is also similar to that presented in reference 28 and is not part of the results obtained in this work.
Response: The scheme was removed. Instead, we used equations of chemical reactions to show the main processes occurring in the course of CO2 hydrogenation, including Fischer-Tropsch synthesis and methanation, as well as methanol formation.
Reviewer #3: The authors should improve the discussion of the results. There are only three references in the whole section and, besides, it is the same sentence as in the introduction section.
Response: First, we performed additional physicochemical characterization of the catalysts, including the catalyst Fe/alumina before and after the reaction. Second, we extended the discussion by considering the results of physicochemical studies. See pages 4-7, highlighted text.
Reviewer #3: The authors include three tables, two of them for alumina supported catalysts and one for Fe/C catalyst, but they do not explain the data shown in table 3 for Fe/C material. Can the authors explain the reason to include this data? They could include a section to compare Fe supports.
Response: We revised tables significantly. Tables 2 and 3 were merged into one table.
Reviewer #3: Figure 1 and Table 2 contain the same data, which is redundant and should be reorganized.
Response: We replaced the figure in order to provide the possibility of comparison of the tested catalysts depending on the reaction temperature.
Reviewer #3: In order to explain the species involved in the process, the include only one TEM image of Fe/C that, theoretically, it is not the catalyst of interest in this contribution. Could the authors include the images for alumina supported catalysts?
Response: We have focused on Fe/Al2O3 catalyst and studied this catalyst using available physicochemical methods. Characterization data have been added in the manuscript (highlighted text in pages 4-7), including TEM, SEM, EDX, TPR, XPS, and DRIFTS with CO used as a probe molecule. Comparison of the Fe/alumina samples before and after catalysis was made using DRIFTS data with CO as a probe.
Reviewer #3: Characterization techniques could be explained before providing the data obtained in the experimental section.
Response: The structure of the manuscripts in the MDPI journals is organized as follows: Results and Discussion should be placed after Introduction, whereas Materials and Methods should be placed after Results and Discussion.
Reviewer 4 Report
The paper deals with the development of a highly selective catalyst towards methanation of CO2. The paper is of interest, especially related to the supercritical conditions, able to increase by an order of magnitude using the supercritical conditions as compared to the gas-phase reactions. I suggest acceptance upon the authors addressing the following minor point:
a) in order to broaden readership’s interest, I suggest to cite relevant literature in the field of homogeneous catalysis applied to the synthesis of relevant organic compounds and polymers: doi: 10.1021/jacs.7b03412; doi: 10.1002/pola.28532; doi: 10.3390/molecules22010021
Author Response
Manuscript ID: molecules-1150813
Title: Carbon Dioxide Reduction with Hydrogen on Fe, Co Supported Alumina and Carbon Catalysts under Supercritical Conditions
Dear Editor and Reviewers,
We would like to thank you for the careful reading of our manuscript and your questions, remarks, and recommendations. We have revised and expanded the manuscript and we kindly ask you to review the revised version. Please find below our responses to the reviewers comments. All significant changes in the manuscript are highlighted in yellow. Also, we checked the use of English and made corrections in the text (not shown in the revised manuscript).
Responses to Reviewer #4 Comments
Reviewer #4: The paper deals with the development of a highly selective catalyst towards methanation of CO2. The paper is of interest, especially related to the supercritical conditions, able to increase by an order of magnitude using the supercritical conditions as compared to the gas-phase reactions. I suggest acceptance upon the authors addressing the following minor point:
- a) in order to broaden readership’s interest, I suggest to cite relevant literature in the field of homogeneous catalysis applied to the synthesis of relevant organic compounds and polymers: doi: 10.1021/jacs.7b03412; doi: 10.1002/pola.28532; doi: 10.3390/molecules22010021
Response:
The reviewer proposed to cite the following papers:
- Doi: 10.1021/jacs.7b03412
Title: Domino Direct Arylation and Cross-Aldol for Rapid Construction of Extended Polycyclic π-Scaffolds
- Doi: 10.1002/pola.28532
Title: Donor–acceptor conjugated copolymers incorporating tetrafluorobenzene as the π‐electron deficient unit
- Doi: 10.3390/molecules22010021
Title: Direct Arylation Strategies in the Synthesis of π-Extended Monomers for Organic Polymeric Solar Cells
We are ready to include publications on homogeneous catalysis relevant to CO2 hydrogenation or any type of CO2 conversion (at least). However, the proposed papers are too far from the topic of our paper.
Round 2
Reviewer 1 Report
I thank the authors for addressing some of my comments. The results are now improved, but the quality of the manuscript submitted is still below standards for publication.
Examples are Figures 4 and 7 and their captions, which show carelessness in writing and finalizing the paper.
The authors should make sure the captions are self-explanatory.
Figure 6 is also not clear: where are the iron particles?
A more detailed discussion of how the physicochemical results relate to the observed activity should be introduced.
I think the manuscript can be considered for publication in Molecules after such minor revisions.
Author Response
Reviewer 1 comments
I thank the authors for addressing some of my comments. The results are now improved, but the quality of the manuscript submitted is still below standards for publication.
Examples are Figures 4 and 7 and their captions, which show carelessness in writing and finalizing the paper.
The authors should make sure the captions are self-explanatory.
Figure 6 is also not clear: where are the iron particles?
A more detailed discussion of how the physicochemical results relate to the observed activity should be introduced.
I think the manuscript can be considered for publication in Molecules after such minor revisions.
Response: We appreciate the additional comments and advices for the improvement of the manuscript of our paper.
We added additional details to the captions of Figs. 4 and 7.
Concerning Fig. 6: it is not iron but iron Fe(3+/Fe2+) oxide that is present at the catalyst surface. It is manifested as darker spots compared to alumina carrier.
The discussion of the physicochemical data in relation to the catalytic performance is extended (page 7, yellow highlighted text).
Reviewer 2 Report
the revised version of the manuscript deserves publication
Author Response
We wish to thank the reviewer for the consideration of our manuscript
Reviewer 3 Report
The authors have considerably modified the manuscript, improving the characterization and the discussion. However, there are still several issues that should be improved for its publication.
- In the first sentence of the abstract the authors claim: “Reduction of CO2 with hydrogen into CO was studied for the first time on alumina-supported Co and Fe catalysts under supercritical conditions.” The authors should clarify the aim of the hydrogenation, because sometimes they indicate that it is the production of CO and in others the study of the CO2 hydrogenation process to CO, CH4 and CH3
- Revise the whole manuscript in order to improve grammar or typo errors. For example, line 44 “only product a RuNi/ZrO2” or the space in line 62.
- In the introduction, only Al2O3 supported catalysts are discussed. Although the data on Fe/C are given in the table and in Figures 1 and 2, it did not provide information or comparison in the rest of the manuscript and in the conclusions. Therefore, I consider it could be removed from the manuscript and be only focused on Al2O3 supported catalysts.
- On the other hand, in line 147 the authors indicate: “Since Fe/Al2O3 catalyst demonstrated the best performance in CO formation compared to Co/Al2O3 and Fe/C catalysts, this catalyst was studied in more detail by using diverse physicochemical methods of characterization.” Nevertheless, in Table 1, higher values of conversion and CO selectivity can be observed for Fe/C catalysts than the values for Fe/Al2O3. Maybe, the authors understand that improvement in terms of lower reaction temperatures, but some clarification is required on this point.
- Figures 5, 6 and 7 are descriptive. They should be more justified and explained compared to the literature. What are the differences between the blue and red lines in Fig 5 due to?
- The conclusion is very short, and the first sentence is not the conclusion of the work, it is the starting point. It should be more detailed.
Author Response
Reviewer 3 comments
The authors have considerably modified the manuscript, improving the characterization and the discussion. However, there are still several issues that should be improved for its publication.
In the first sentence of the abstract the authors claim: “Reduction of CO2 with hydrogen into CO was studied for the first time on alumina-supported Co and Fe catalysts under supercritical conditions.” The authors should clarify the aim of the hydrogenation, because sometimes they indicate that it is the production of CO and in others the study of the CO2 hydrogenation process to CO, CH4 and CH3
Revise the whole manuscript in order to improve grammar or typo errors. For example, line 44 “only product a RuNi/ZrO2” or the space in line 62.
In the introduction, only Al2O3 supported catalysts are discussed. Although the data on Fe/C are given in the table and in Figures 1 and 2, it did not provide information or comparison in the rest of the manuscript and in the conclusions. Therefore, I consider it could be removed from the manuscript and be only focused on Al2O3 supported catalysts.
On the other hand, in line 147 the authors indicate: “Since Fe/Al2O3 catalyst demonstrated the best performance in CO formation compared to Co/Al2O3 and Fe/C catalysts, this catalyst was studied in more detail by using diverse physicochemical methods of characterization.” Nevertheless, in Table 1, higher values of conversion and CO selectivity can be observed for Fe/C catalysts than the values for Fe/Al2O3. Maybe, the authors understand that improvement in terms of lower reaction temperatures, but some clarification is required on this point.
Figures 5, 6 and 7 are descriptive. They should be more justified and explained compared to the literature. What are the differences between the blue and red lines in Fig 5 due to?
The conclusion is very short, and the first sentence is not the conclusion of the work, it is the starting point. It should be more detailed.
Response: We appreciate the additional comments and advices for the improvement of the manuscript of our paper. We revised the abstract and added explanations related to the products of the CO2 hydrogenation reaction (highlighted in yellow).
The grammar of the manuscript was double-checked and the typos were corrected.
Alumina-based systems were chosen as the main objects of the study, but it was interesting to see whether or not the replacement of alumina for the neutral (inert) carrier such as alumina affects the conversion/selectivity pattern. Therefore, we would prefer to keep the results for Fe/C catalyst studied for comparative purposes. We revised the discussion and comparison on pages 3,4 in order to explain why we compare alumina and carbon as carriers (highlighted in yellow).
Also, we extended the conclusions and added some discussion of the IR spectra of Fe-alumina catalyst before and after catalysis.